# Nano-Delivery System of Ethanolic Extract of Propolis Targeting *Mycobacterium tuberculosis* via Aptamer-Modified-Niosomes

**DOI:** 10.3390/nano13020269

**Published:** 2023-01-08

**Authors:** Sirikwan Sangboonruang, Natthawat Semakul, Sureeporn Suriyaprom, Kuntida Kitidee, Jiaranai Khantipongse, Sorasak Intorasoot, Chayada Sitthidet Tharinjaroen, Usanee Wattananandkul, Bordin Butr-Indr, Ponrut Phunpae, Khajornsak Tragoolpua

**Affiliations:** 1Division of Clinical Microbiology, Department of Medical Technology, Faculty of Associated Medical Sciences, Chiang Mai University, Chiang Mai 50200, Thailand; 2Infectious Diseases Research Unit (IDRU), Faculty of Associated Medical Sciences, Chiang Mai University, Chiang Mai 50200, Thailand; 3Department of Chemistry, Faculty of Sciences, Chiang Mai University, Chiang Mai 50200, Thailand; 4Department of Biology, Faculty of Sciences, Chiang Mai University, Chiang Mai 50200, Thailand; 5Center for Research and Innovation, Faculty of Medical Technology, Mahidol University, Salaya, Nakhon Pathom 73170, Thailand; 6Office of Disease Prevention and Control 1, Chiang Mai 50200, Thailand

**Keywords:** niosomes, nano-delivery, propolis, Ag85A aptamer, tuberculosis

## Abstract

Tuberculosis (TB) therapy requires long-course multidrug regimens leading to the emergence of drug-resistant TB and increased public health burden worldwide. As the treatment strategy is more challenging, seeking a potent non-antibiotic agent has been raised. Propolis serve as a natural source of bioactive molecules. It has been evidenced to eliminate various microbial pathogens including *Mycobacterium tuberculosis* (Mtb). In this study, we fabricated the niosome-based drug delivery platform for ethanolic extract of propolis (EEP) using thin film hydration method with Ag85A aptamer surface modification (Apt-PEGNio/EEP) to target Mtb. Physicochemical characterization of PEGNio/EEP indicated approximately −20 mV of zeta potential, 180 nm of spherical nanoparticles, 80% of entrapment efficiency, and the sustained release profile. The Apt-PEGNio/EEP and PEGNio/EEP showed no difference in these characteristics. The chemical composition in the nanostructure was confirmed by Fourier transform infrared spectrometry. Apt-PEGNio/EEP showed specific binding to *Mycobacterium* expressing Ag85 membrane-bound protein by confocal laser scanning microscope. It strongly inhibited Mtb in vitro and exhibited non-toxicity on alveolar macrophages. These findings indicate that the Apt-PEGNio/EEP acts as an antimycobacterial nanoparticle and might be a promising innovative targeted treatment. Further application of this smart nano-delivery system will lead to effective TB management.

## 1. Introduction

Tuberculosis (TB), principally caused by *Mycobacterium tuberculosis* complex (Mtbc), is one of the pernicious airborne diseases and has been a major global health concern. According to recent reports, the number of new cases of TB has risen to over a million, with approximately half a million of these cases attributed to drug-resistance development and more than 1.4 million deaths annually [1,2,3]. TB infection with *M. tuberculosis* (Mtb) occurs mostly in the lungs through inhaled contaminated aerosols and residences within the alveolar macrophage facilitating the escape of Mtb from the immune response, resulting in unsuccessful treatment [4]. Currently, conventional TB treatment relies on the combination of various drugs over a lengthy period, generally of at least 6 months, that cause life-threatening side effects and drug resistance development [1]. According to these issues, identifying a non-antibiotic agent to counter this deadly pathogen might be helpful. 

For centuries, various natural extracts have been used for bactericidal purposes. Propolis, a resinous material collected by bees is a potent natural substance that provides an abundant source of bioactive compounds and has been suggested for the development of new drugs [5,6]. The main chemical groups in propolis include phenolic acids or their ester flavonoids, terpenes, aromatic aldehydes and alcohols, fatty acids, stilbenes and beta-steroids [7]. Several works have investigated and reported on its properties in anti-cancer, antioxidant, immunomodulatory, anti-inflammatory, and anti-microorganisms [5,7,8]. Generally, the anti-bacterial mechanism of propolis is based on the function of its polyphenolic constituents that possibly affects the cytoplasmic membrane function, nucleic acid synthesis, metabolism, or formation of biofilm [9]. For the treatment of TB, many studies have previously demonstrated the growth inhibition of propolis extracts toward Mtb [9,10] and its synergistic effects with various antitubercular drugs such as isoniazid, rifampicin, and streptomycin [11]. These strengthen the effectiveness of propolis as a candidate therapeutic substance for Mtb infection.

Nanoparticle-based drug delivery is an effective method with which to bridge the gap in research related to the lack of solubility and ability of bioactive compounds including propolis to reach targets. Amongst nano-carrier systems, lipid-based formulations provide unique advantages to penetrate to the mucus layer [4], fuse with bacterial membrane and enhance cellular uptake in macrophages [12]. Non-ionic surfactant vesicles (niosomes), a class of lipid vesicles are a self-assembly of non-ionic surfactants and cholesterol [13] that might be compatible with the lipid-rich structure of Mtb cell wall components [14]. Previously, the formulation of niosome was produced to encapsulate ethanolic extract of propolis (EEP) against *Staphylococcus aureus* and *Candida albicans* [15]. In this study, the therapeutic properties of EEP for the purpose of anti-Mtb activity were investigated through the niosome system.

To increase the efficacy of drug delivery to the target pathogen, the surface of niosomes was designed to contain the functional groups that chemically interact with the trafficking of specific molecules to the target site. Recently, the application of aptamer (Apt), short single-stranded DNA or RNA oligonucleotides has been useful in various fields [16]. Apt can fold into three-dimensional structures and specifically bind to its target with high affinity [17,18]. Similar to monoclonal antibodies, Apt has been utilized for target recognition and applied for diagnostics, biosensors, and therapeutics purposes [16]. The generation and application of Apt against biomolecules of Mtb was also previously reported, such as MPT64 [19], CFP-10 and ESAT-6 [20], alongside Ag85A [21] for the detection of Mtb. 

In this present study, the EEP-loaded PEGylated niosome (PEGNio/EEP) was fabricated and Ag85A Apt conjugate was attached to PEGNio/EEP (Apt-PEGNio/EEP) via an amide coupling reaction. The obtained nano-formulation was subsequently evaluated for the specific binding to Mtb and anti-Mtb activity in vitro. Thus, we hypothesized that the Mtb surface could be recognized with an Apt functionalized niosome, facilitating EEP release, which directly affected the target pathogen, and Mtb was ultimately eradicated. 

## 2. Materials and Methods

### 2.1. Materials

Ag85A Apt was synthesized according to Ansari et al., 2017 [21] and some modification of Ag85A Apt with amine (5′-GCT GTG TGA CTC CTG CAA GCG GGA AGA GGG TAA GGG GAG GGA GGG TAA CGC GGA GAA GGC AAG CAG CTG TAT CTT GTC TCC/NH_2_/-3′) and Cy5 (5′-/Cy5/GCT GTG TGA CTC CTG CAA GCG GGA AGA GGG TAA GGG GAG GGA GGG TAA CGC GGA GAA GGC AAG CAG CTG TAT CTT GTC TCC/NH_2_/-3′) were purchased from Integrated DNA Technologies (Coralville, IA, USA). Sorbitan monostearate (Span 60), cholesterol (CHOL) and N-hydroxysulfosuccinimide sodium salt (NHS) were purchased from Sigma-Aldrich (St Louis, MO, USA). 1,2-distearoyl-sn-glycero-3-phosphoethanolamine-N-(polyethylene glycol) (2000) carboxylic acid (DSPE-PEG-COOH) was purchased from Avanti (Birmingham, AL, USA) and 1-ethyl-3-(3-dimethylaminopropyl) carbodiimide hydrochloride (EDC) was purchased from Thermo Scientific (Waltham, MA, USA). EEP powder was kindly provided by Bee Product Industry Co., Ltd., Lamphun, Thailand.

### 2.2. Bacterial and Cell Culture

Mtb H37Rv and nontuberculous Mycobacteria (NTM), including *M. avium*, *M. intracellulare*, *M. kansasii*, and *M. phlei*, were cultured in Middlebrook 7H9 (M7H9) medium supplemented with 10% oleic albumin dextrose catalase (OADC). *Staphylococcus aureus* and *Escherichia coli* were cultured in brain-heart infusion (BHI) broth. *Candida albicans* was cultured in Sabouraud dextrose broth (SDB). The broth culture was adjusted to the 0.5 McFarland for further experiment. The protocol was reviewed and approved by the Institutional Biosafety Committee, Chiang Mai University (permit number: CMUIBC-0565004).

Alveolar macrophage cell (NR8383) was purchased from the American Type Culture Collection (ATCC, Manassas, VA, USA) and cultured in Kaighn’s modification of Ham’s F-12 medium (F-12K) (supplemented with 15% fetal bovine serum (FBS), penicillin (100 units/mL) and streptomycin (100 µg/mL)) and maintained in a humidified atmosphere of 5% CO_2_ at 37 °C.

### 2.3. Preparation of EEP Solution

The ethanol solution of EEP was prepared from 100 mg of EEP powder dissolved in 1 mL of ethanol with shaking for 20 min at room temperature. The supernatant of EEP was obtained after centrifugation at 1000× *g* for 15 min [22] and further subjected to measure total phenolic content as gallic acid equivalents (GAE).

### 2.4. Assessment of Total Phenolic Content

Total phenolic content of EEP was quantified according to the Folin–Ciocalteu method [23]. In brief, the sample (100 µL) was mixed with 50 µL of 50% Folin–Ciocalteu reagent and 500 µL of distilled water and 100 µL of ethanol. Then, the mixtures were left standing in the dark at room temperature for 5 min. The 100 µL of 5% *w/v* sodium carbonate solution was added and incubated in the dark room temperature for 1 h. Then, 200 µL of reaction mixture were added into the 96-well plate and the absorbance was measured at 725 nm using a spectrophotometer. The total phenolic content was calculated through the gallic acid calibration curve. The results were expressed as mg GAE/mL.

### 2.5. Optimization of Nano-Formulations

The nano-formulations were modified and prepared by the thin film hydration method [24] using different molar ratios of non-ionic surfactant and CHOL, as indicated in Table 1. Briefly, Span 60, CHOL and DSPE-PEG-COOH were dissolved in 2 mL of chloroform and subsequently added with 0.5 mL of ethanol solution of EEP in a round-bottomed flask. The solvent was evaporated using a rotary evaporator at 60 °C with constant rotation under reduced pressure to form a lipid thin film. Then, the thin film was hydrated with PBS to obtain PEGNio/EEP at 60 °C for 60 min. For preparing empty niosomes (PEGNio), the ethanol without EEP was performed. The particle size of formulations was reduced using an ultrasonic probe homogenizer (Hielscher UP50H; Hielscher, NJ, USA) at 80% amplitude for 10 min in an ice bath. The obtained formulations were further subjected for physicochemical characterization and entrapment efficiency (EE).

### 2.6. Conjugation of Apt to PEGNio/EEP

The functionalized-Apt-PEGNio/EEP was achieved through an amide linkage between a carboxyl group on the PEG of NPs and amide-modified Apt via an EDC/NHS reaction [25]. Briefly, the appropriate amount of PEGNio/EEP was incubated with 100 μL of 40 mM EDC and 100 μL of 10 mM NHS for 15 min at room temperature with gentle stirring. Then, the NHS-activated particles were covalently linked to 50 μL of 1 μg/mL 3′-NH_2_ Apt for 1 or 2 h at room temperature. The resulting functionalized PEGNio/EEP was washed and preserved with DNase-RNase free water. The conjugated structure of Cy5 Apt-PEGNio/EEP was evaluated based on fluorescent intensity of Cy5 by multimode microplate reader (CLARIOstar with LVF, BMG LABTECH, Ortenberg, Germany). Furthermore, the amide coupling was also confirmed by FT-IR analysis. Schematic representation of the bioconjugation process is shown in Figure 1. 

### 2.7. Physicochemical Characterization

#### 2.7.1. Size Distribution and Zeta-Potential (ZP)

The average particle size, polydispersity index (PDI), and ZP of formulations were determined using dynamic light scattering (DLS) with Zetasizer (Malvern Instruments, Worcestershire, UK). Each sample was diluted to suitable concentration and measured based on at least three measurements in three individual runs.

#### 2.7.2. Entrapment Efficiency (EE)

Free EEP was separated from PEGNio by ultracentrifugation with 10 kDa molecular weight cut-off at 8000× *g*, 4 °C, for 2 h. The pellets were re-suspended in PBS and lysed in isopropanol. The EE of EEP in PEGNio was estimated by total phenolic content assay. The following equation was used to determine the %EE of EEP.
(1)$$\%\mathrm{EE}=\frac{\mathrm{Total}\;\mathrm{phenolic}\;\mathrm{content}\;\mathrm{of}\;\mathrm{EEP}\;\mathrm{in}\;\mathrm{formulation}}{\mathrm{Total}\;\mathrm{phenolic}\;\mathrm{content}\;\mathrm{of}\;\mathrm{EEP}\;\mathrm{used}\;\mathrm{in}\;\mathrm{preparation}}\times 100$$

#### 2.7.3. Morphology of NPs

The morphological characteristics of nano-formulations were observed by scanning transmission electron microscopy (STEM). A drop of niosome sample was stratified onto a carbon coated copper grid and then stained with 1% phosphotungstic acid. After absorption of the sample on the grid, the morphology was observed by JSM-IT800 Ultrahigh Resolution Field Emission SEM (JEOL, Peabody, MA, USA).

#### 2.7.4. Fourier Transform Infrared (FT-IR) Spectrometry

To ensure that the nano-formulations were successfully prepared, the functional group of chemical substances in the formulations was studied using FT-IR spectrometry. The lyophilized samples were measured using attenuated total reflectance (ATR) mode of Bruker Tensor 27 FT-IR spectrometer (Billerica, MA, USA). FT-IR was recorded in the range of 4000–400 cm^−1^ at a resolution of 2 cm^−1^ and compared with the spectra of precursors. 

### 2.8. In Vitro Release Profile

Drug release assay was performed to assess the amount of EEP released from the formulations in modified stimulated lung fluid (mSLF) solution [26], adjusted to pH 7.4 and pH 5.5, in accordant with physiological conditions and an acidic environment that reflects intracellular infection, respectively. The samples were rotated, and the supernatant was collected by centrifugation at specific time intervals, followed by the replacement with the same volume of fresh mSLF solution. The mSLF samples were assessed for the total phenolic content. Each experiment was repeated in triplicate in three individual experiments.

### 2.9. Examination of an Interaction between Apt-PEGNio/EEP and Mtb 

To study the targeting specificity of Apt-PEGNio/EEP, binding of Cy5 Apt-PEGNio/EEP was investigated in Mtb, NTM, *S. aureus*, *E. coli*, and *C. albicans* using confocal laser scanning microscope (CLSM). In brief, freshly culture of selected microorganisms was fixed with 4% paraformaldehyde and incubated with Cy5 Apt-PEGNio/EEP for 1 h. Then, mycobacterial cell wall and nucleic acid of *S. aureus*, *E. coli* and *C. albicans* were stained with Auramine O and DAPI, respectively, for 15 min. After washing with PBS, the bacterial suspension was dropped on glass slide and observed using CLSM (FluoView FV1000; Olympus Optical, Tokyo, Japan and Nikon AX; Nikon Instruments Inc., Melville, NY, USA).

### 2.10. Assessment of Anti-Mtb Activity by Resazurin Microtitre Assay (REMA) [27]

Two-fold serial dilutions of EEP, PEGNio, PEGNio/EEP, or Apt-PEGNio/EEP [preserved in sterile PBS containing vancomycin (50 μg/mL) and gentamicin (50 μg/mL)] were diluted in M7H9 medium and added (100 μL) into 96-well plate. An inoculum of Mtb was adjusted in medium and 100 μL of bacterial suspension was added per well (5 × 10^4^ CFU). Growth controls containing no EEP and NPs, and a sterile control without bacteria were also prepared for each assay. The plate was incubated at 35 °C for 7 days before adding 30 μL of 0.01% resazurin. After 24 h, the Mtb viability was determined based on a change in color from blue (oxidize state) to pink (reduced state) at 560 nm.

### 2.11. Cytotoxicity Test

The effect of the formulations on cell viability was investigated on NR8383 cells by REMA assay, according to Santos et al., 2018 [28] with some modification. Briefly, NR8383 (5 × 10^5^ cells/well) was seeded in a 96-well tissue culture plate and cultured for 24 h. The cells were treated with various concentrations of PEGNio, PEGNio/EEP or Apt-PEGNio/EEP (preserved in sterile PBS containing vancomycin (50 μg/mL) and gentamicin (50 μg/mL). After 24 h incubation, cell viability was determined after adding 20 μL of 0.01% resazurin and incubated for 24 h. The change in color of resazurin from blue to pink was then measured at 560 nm.

### 2.12. Statistical Analysis

All results are represented as mean ± standard error of the mean (SEM). Comparisons between groups were made using one-way analysis of variance. Notably, *p* < 0.05 was considered statistically significant. All calculations were performed using SPSS Statistics 23.0 software (IBM Corp., Armonk, NY, USA). 

## 3. Results

### 3.1. Anti-Mtb Activity of EEP

The activity of EEP against Mtb growth was investigated by REMA assay. The viability of Mtb was indicated by a change in color from blue (oxidize state) to pink (reduced state). The positive control for Mtb growth inhibition was performed using isoniazid (INH) at 1 μg/mL (Figure 2a). The result showed that the EEP significantly exhibited dose-dependent decrease in mycobacterial cell viability, whereas the presence of vehicle control (VC) did not notably induce growth inhibition when compared to the growth control (Control) (Figure 2b). The 50% growth inhibition concentration (IC_50_) of EEP was calculated and correlated to 30 μg/mL GAE total phenolic contents. This result clearly revealed an anti-Mtb activity of EEP.

### 3.2. Physicochemical Characteristics 

The nano-formulations were prepared from different ratios of non-ionic surfactant and CHOL. The physicochemical properties and morphology of PEGNio and PEGNio/EEP are shown in Table 2 and Figure 3a,b. The mean particle sizes of F2 and F3 were approximately 150–190 and 140–320 nm, respectively, with a PdI of 0.2–0.3 indicating a homogeneous size distribution, whereas the F1 showed an increase in size ranging from 210–280 nm with a high polydispersity from 0.4 to 0.5. All formulations showed negative ZP ranging from −15 to −35 mV. Moreover, the F2 formulation had the highest level of EEP, approximately 80% with total phenolic content 0.14 ± 0.016 mg/mL GAE, while the F1 and F3 were reached 39% and 42%, respectively. From these results, the F2 formulation offered the higher properties to be employed for further bioconjugation with Apt. The morphological characterization of the F2 was then observed. The TEM images of the F2 showed a relatively uniform nanoparticle size with spherical shape that concordant with DLS analysis (Figure 3a,b). 

The functional groups of PEGNio and PEGNio/EEP were further studied by FT-IR spectroscopy. The FT-IR spectra of Span 60, CHOL, DSPE-PEG-COOH, and freeze-dried PEGNio are shown in Figure 3c. The spectrum of Span 60 showed peaks at 1727 cm^−1^ (C=O stretching), 1168 cm^−1^ (-CO_2_-), and 2921 and 2846 cm^−1^ (asymmetric and symmetric C-H stretching). The spectrum of CHOL displayed several peaks, such as 1047 cm^−1^ (C–O stretching), 2800–2900 cm^−1^ (C-H stretching), and a broad peak at 3100–3600 cm^−1^ (O-H stretching). The spectrum of DSPE-PEG-COOH showed peaks at 3450 cm^−1^ (O-H stretching) and 1737 cm^−1^ (C=O stretching) of the carboxylic acid (-COOH) group. It was found that the characteristic peaks of Span 60, CHOL and DSPE-PEG-COOH were presented in the FT-IR spectrum of the PEGNio sample. The encapsulation of EEP in PEGNio was further studied by FT-IR analysis, as shown in Figure 3d. FT-IR spectra of EEP showed peaks at 3357 cm^−1^ (O-H stretching) and 1633 cm^−1^ (C=O stretching) of phenolic compounds in EEP. In PEGNio/EEP, both sets of peaks of PEGNio and EEP appeared in FT-IR spectra of PEGNio/EEP sample, suggesting EEP was successfully incorporated into PEGNio.

### 3.3. Conjugation and Characterization of Apt-PEGNio/EEP

As the conjugation of Apt to PEGNio/EEP required the formation of covalent couple between the -COOH of PEG on the NPs surface and the 3′-NH_2_ Apt, the conjugation relying on fluorescent intensity of Cy5-labeled Apt was optimized. As shown in Figure 4a,b, the presence of Cy5 in the formulations showed a significant fluorescent intensity (*p* < 0.05), when compared to unconjugated NPs. This result supported the reaction between Cy5-labeled Apt and NPs was achieved. Additionally, the reaction time was not significantly different either 1 or 2 h. Thus, the reaction time at 1 h was chosen to conduct Apt-PEGNio/EEP in further experiments.

The appearance and characteristics of decorated PEGNio or PEGNio/EEP with Apt was evidently demonstrated in Figure 4c. Similar to the previous results, the average size, PDI and ZP of Apt-PEGNio/EEP slightly differed from the unmodified particles. The particle characterization after coupling to Apt was increased in size to 198.30 ± 5.02 nm, while the PDI and ZP values were found at 0.36 ± 0.02 and −19.55 ± 2.27 mV, respectively. 

In addition, the successful conjugation of Apt-NH_2_ onto PEGNio and PEGNio/EEP was confirmed through FT-IR analysis. As shown in Figure 4d, the FT-IR spectrum of PEGNio exhibits the characteristic peaks corresponding to unmodified DSPE-PEG-COOH with additional functional absorption peaks of Span 60 and CHOL which was described vide infra. FT-IR spectrum of Apt-NH_2_ shows a peak at 3515 cm^−1^_,_ which can be assigned as N-H stretching of the amino group. The conjugated sample showed a broad and strong signal at 3342 cm^−1^, which is attributed to the successful formation of an amide bond. In addition, the absorption at 1647 cm^−1^, ascribed as C=O stretching of amide, indicates successful peptide coupling. Interestingly, the applicability of the Apt-NH_2_ conjugation with PEGNio/EEP was demonstrated. We directly employed PEGNio/EEP precursor to conjugate with Apt-NH_2_ precursors in the presence of NHS/EDC. FT-IR spectrum of the resultant Apt-PEGNio/EEP sample contains peaks similar to Apt-PEGNio, as shown in Figure 4e. This result demonstrates that NHS/EDC mediated-amide bond formation can be universally applied for the conjugation of PEGNio and the post-encapsulated conjugation of PEGNio/EEP with Apt-NH_2_.

### 3.4. Binding Ability of Apt-PEGNio/EEP to Mtb

The ability of Apt-PEGNio/EEP to recognize Ag85A-membrane-bound protein was verified with Mtb, NTM included *M. avium*, *M. intracellulare*, *M. kansasii* and *M. phlei* as well as other microorganisms, such as *S. aureus*, *E. coli* and *C. albicans*. As shown in Figure 5a,b, the Cy5 conjugated Apt-PEGNio/EEP (red) accumulated surrounding Mtb, including *M. avium*, *M. intracellulare*, and *M. kansasii*, that stained their cell wall with Auramine O (green), whereas an intensity of Cy5 was found to be low in *M. phlei*. However, binding of the NPs did not show in *S. aureus*, *E. coli* and *C. albicans,* which were indicated by DAPI labeling nucleic acid (blue). Further observation of Apt-PEGNio/EEP distribution on the target Mtb was performed. It was demonstrated that the Cy5 Apt-PEGNio/EEP was intensely gathered on the Mtb cell wall as shown in Figure 5c. This result implied that the Apt-PEGNio/EEP had a selective binding activity to Mtb and some mycobacterium.

### 3.5. In Vitro Release of Apt-PEGNio/EEP

The in vitro release profile of EEP from the Apt-PEGNio/EEP was determined during an experiment of 48 h in mSLF at pH 7.4 and 5.5 in a corresponding physiological condition and acidic environment, respectively. The result showed no difference for release rate and cumulative release amount among release profiles between the two conditions (Figure 6). The initial burst release of EEP was approximately 30–40% at the first 3 h. The remaining EEP was then released gradually over the following hours and more than 60% during 48 h. This result suggested the capability of the niosome system to release EEP in a sustained profile. 

### 3.6. Anti-Mtb Activity and Cytotoxicity of Apt-PEGNio/EEP

Even though, the study of Apt-PEGNio/EEP and Mtb binding clearly demonstrated the accumulation of the functionalized-NPs on the Mtb cell wall, however, their anti-Mtb effect were essentially determined. Dosage of each nano-formulations were correlated to the total phenolic content of EEP and employed to treat Mtb cells. As demonstrated in Figure 7a, the activities of PEGNio/EEP and Apt-PEGNio/EEP were statistically significant to inhibit Mtb growth, as compared to the PEGNio (* *p* < 0.05). The reduction of Mtb viability induced by Apt-PEGNio/EEP required a lower dosage than non-functionalized formulation (*p* < 0.05). This result suggested that the Apt-functionalized-PEGNio/EEP could not only increase target specificity but also enhance therapeutic activity of EEP superior to unmodified system.

For further application in pulmonary drug delivery, the cytotoxicity of PEGNio, PEGNio/EEP, and Apt-PEGNio/EEP were preliminary investigated in NR8383 cell line. The number of particles used in this study ranged from 10^10^–10^11^ particles that corresponded to the concentration of total phenolic content of EEP between 0–16 μg/mL. The result demonstrated nontoxicity of all formulations to the cells with relative cell viabilities above 80% after administration for 24 h (Figure 7b). 

## 4. Discussion

The dramatic increase in TB cases and emergence of drug-resistant TB has made the management and medication of this disease more complicated. Moreover, TB patients suffer from adverse effects and costly treatment by conventional therapeutic agents [3]. Thus, identifying an alternative source of medicine is a promising solution that is urgently needed. The screening of antimicrobial substances from natural sources is ongoing and the research has revealed many potent compounds including propolis. Propolis is considered as a candidate for therapeutic use due to being rich in functional bioactive substances and its reported safety profile in animal models and humans [1,6,8]. The anti-mycobacterial activity of propolis was previously demonstrated against Mtb and NTM. In previous reports, EEP was shown to inhibit the growth of the Mtb H37Rv reference strain [29] and showed a synergistic effect with an antitubercular drug against mycobacteria obtained from clinical isolates [11]. It was also shown that the ethanolic extract of green propolis had an inhibitory effect on *M. chelonae* and *M. kansasii* [30]. Sawicki et al., 2022 also demonstrated the bacteriostatic effect of *Apis mellifera* L. EEP and inhibitory effect of *Trigona* sp. EEP against Mtb H37Ra [9]. By using the molecular docking technique, various known constituents in propolis, such as flavonoids, terpenoids, simple phenolics, pterocarpan, phenylethanoid derivative, and stilbenes, were found to interact with the key proteins related to Mtb growth and many essential mycobacterial pathways such as biosynthesis of cell wall and co-factors, and signal transduction [10]. The possible mechanism of antibacterial activity of propolis is related to the action of its individual polyphenolic constituents of plant and geographic origins as well as bee species [9,31,32]. Thus, the anti-Mtb activity of EEP was investigated to verify the effectiveness of bioactive substances derived from EEP as new therapeutic agents. From our results, EEP significantly exhibited anti-microbial activity against Mtb H37Rv at 50% growth inhibition concentration, which correlated to 30 μg/mL GAE total phenolic contents. 

Nevertheless, therapeutic use of EEP for TB through direct administration might not be practical in many aspects. First, a major limitation is the lack of water solubility of EEP [33]. Second, the unique lipid-rich structure of Mtb serves as an impermeable barrier to insidious substances and drugs, which could result in treatment failure [34]. Third, Mtb is considered as an intracellular pathogen that usually persists in macrophages and is surrounded by lung surfactant [14]. Thus, the lack of accessibility of anti-microbial substances to reach the target site remains a considerable challenge. 

The use of a nano-drug delivery system is regarded as a useful strategy to entrap and deliver the payload to reach the target site [35]. According to the special lipid-rich structure of Mtb cell wall and lung surfactant, lipid-based nanoparticles might be an attractive candidate platform. Niosome or non-ionic surfactant vesicles is a lipid-based carrier produced from the self-assembly of non-ionic surfactants and lipid [13]. Previous studies revealed many useful characteristics of niosome to improve the therapeutic efficiency of various kind of functional molecules such as synthetic drugs, proteins, small peptides, nucleic acids as well as natural extracts [13,36]. Additionally, niosomes are able to administer through various routes including subcutaneous, intramuscular, intravenous, ocular, transdermal, and pulmonary deliveries [13]. To date, many anti-TB drugs, such as rifampicin, isoniazid, pyrazinamide and gatifloxacin, have been successfully delivered to the target via niosomal carriers [37,38,39,40]. Therefore, niosome-encapsulated EEP for Mtb treatment was established in this current work. However, achievement of the target pathogen requires navigating specific molecules that are able to selectively and specifically bind to the pathogenic cells. The surface of niosome is tunable with the bioconjugation approach, mostly using PEG. PEG provides the end group for conjugation with various kind of ligands [41]. Furthermore, PEG is a non-toxic, non-immunogenic polymer and has been approved by the FDA for internal use [41,42]. 

In this study, we thus focused on Ag85A, the cell wall protein of Mtb, as the candidate target. The Ag85 proteins are secreted and retained in the cell wall of Mtb [43]. In 2017, Ansari and colleagues firstly investigated the Apt, which specifically binds to Ag85A for Mtb detection [21]. As the folding of the three-dimensional structure, Apt possesses high specificity and affinity to its targets, which might be small molecules, proteins, or pathogens. Moreover, Apt is more stable, and can be more easily synthesized with high purity and non-immunogenicity compared to antibodies [21,44]. Therefore, the Ag85A Apt was employed to decorate the surface of niosome for targeted treatment application. Apt-PEGNio/EEP was constructed to facilitate EEP uptake and increase the therapeutic efficacy of EEP delivery towards the Mtb via the interaction of Apt specific to Ag85A. The desired structure of our nano-formulation was shown in Figure 1. 

Since the formation of bilayer vesicles is mainly based on the chemical structure and hydrophilic-lipophilic balance [13,45], the PEGNio and PEGNio/EEP were initially constructed with the optimization of non-ionic surfactant and lipid components. From the results, it can be seen that all formulations were successfully formed in different sizes of nanoparticles. These results could be explained by the influence of the non-ionic surfactant and lipid ratio that affect the molecular geometry and integrity of the vesicular system [24,46,47]. As the resulting F1 formulation, the particles showed the largest size with a wide range of dispersity. F2 and F3 provided preferable characteristics with reduction in size and PDI. These variations in size might be attributed to the stature between Span 60 and CHOL that correlates with the previous work [15]. It was previously described that a longer alkyl chain of Span 60 can produce a larger particle size, while a decrease in size is influenced by the addition of CHOL. CHOL can help to improve the mechanical strength and rigidity of the vesicle membrane [47]. The presence of CHOL in the formulation resulted in interactive forces between Span 60 and CHOL, and consequently a reduction in the curvature of the vesicles [13,15]. Hence, optimizing material proportion is one of the critical steps to obtain the desired characteristic features of NPs [47]. The EEP-loaded particles presented a larger size than the empty formulation as a result of the bilayer composition properties as well as the interaction or repulsion forces between the bilayers and the entrapped drug [48]. The surface charge of nano-vesicles presented a negative potential of approximately −20 mV. This indicates moderate stability [49], as ZP values outside of the range of −30 mV to +30 mV are needed for the stability of drug delivery systems [48,50]. The efficacy to entrap the EEP was different, as indicated by the %EE and the total phenolic content of EEP. In this study, the highest level of EE was gained with the F2 formulation, of approximately 80%, which is a higher gain in EE than in other previous works [15,51]. This effect might be due to the longer alkyl chain length of Span 60 (Span 60 > Span 40 > Span 20), as the bilayer expansion restricts the EEP in the structure [48]. Moreover, the higher EE level of our formulation is possibly due to the addition of PEG in the system. PEG-modified niosome not only offers a higher EE but also prevents the particles from the reticuloendothelial system (RES) clearance and leads to a longer blood circulation time, thus improving therapeutic efficacy [41,42]. Considering the EE, the F2 formulation was constructed to bioconjugate with Apt for further investigation.

Besides improvement of physical properties, PEG is widely used for bioconjugation application [42]. The addition of PEG to the niosome surface allows various biomolecules, such as oligonucleotides, peptides, growth factors, and antibodies, to attach, leading to the accumulation of the drug on the target site [52]. In this current work, we designed the niosome decorated with Apt specific to Ag85A of Mtb. The conjugation of Apt onto the surface of PEGNio/EEP was achieved through an amide coupling reaction. DSPE-PEG-COOH served as the carboxylic end group to react with the amine-modified Apt in the presence of NHS and EDC [53]. The presence of Apt in the formulation and carboxyl-amine crosslinking of the Apt-PEGNio or Apt-PEGNio/EEP conjugates was verified based on fluorescence assay and FT-IR analysis (Figure 4). The vesicle size after the conjugation increased by ~30 nm, in comparison to the EEP-loaded particle size before coupling to Apt. Corresponding to other works, this result presumably resulted from the increased size of Apt [25,53], while the effect of ZP was less prominent [54]. 

For applications in targeted drug delivery, the functionalized nano-carrier was also verified to have a specific binding ability. From our findings, Apt-PEGNio/EEP selectively attaches to mycobacterium including Mtb, *M. avium*, *M. intracellulare*, and *M. kansasii*, except *M. phlei*, while binding with other bacterial strains (*S. aureus* and *E. coli*) and yeast (*C. albicans*) was not detected (Figure 5). As mentioned earlier, Ag85 is an essential protein associated with the cell wall of Mtb. However, it consists of three distinct variants (Ag85A, B, or C) with varied ratios in response to the environment, usually expressed at a steady-state ratio of 2:3:1, respectively, and serves as a highly conserved protein across the *Mycobacterium* genus [21,43,55,56]. As previously reported, antibodies against Mtb Ag85 proteins showed a broad cross-reaction with Ag85 homologues expressed by other mycobacterial species (*M. bovis*, *M. kansasii*, *M. avium*, *M. xenopi*, *M. gordonae*, *M. fortuitum*, *M. phlei* and *M. smegmatis*) [57]. Thus, the Apt-functionalized niosome possibly possessed cross-interaction with NTM to some extent. However, the absence of Cy5 conjugated Apt-PEGNio/EEP on *M. phlei* can perhaps be attributed to the expression level of Ag85 protein on its surface. As previously mentioned, Ag85A can be found at a lower rate than Ag85B [43] and our earlier result indicates a small amount of secreted Ag85B in *M. phlei* based on ELISA assay (unpublished data). Thus, the expression of Ag85A might be more depleted. Overall, the likelihood of Apt-PEGNio/EEP binding to *M. phlei* was found to be low.

Sustained release is one of the important features of nano drug delivery systems that occurs by means of passive transport through the bilayer membrane [53]. Drug release in a sustained manner aids in prolonging drug activity during the treatment period and reduces side effects associated with drug overdose [58]. In this study, the release of EEP from Apt-PEGNio/EEP was investigated at pH 7.4 and 5.5, in accordance with physiological conditions and to provide an acidic environment that reflects intracellular infection, respectively. With the release profile result, it was clearly demonstrated that Apt-PEGNio/EEP provided a sustained-release of EEP during 48 h in both pH 7.4 and 5.5 conditions. The EEP released from Apt-PEGNio/EEP was divided in two stages. First, we consider an initial rapid phase as a result of free EEP in the outer surface. Second, a slower release phase was attributed to the diffusion of EEP across the bilayer. This suggests that Apt-PEGNio/EEP can function either in the regular extracellular environment or intracellular infection conditions. 

The anti-Mtb activity of Apt-PEGNio/EEP was found to be superior to PEGNio/EEP, suggesting that the Apt modified niosome surface might increase the possibility of the niosome capturing the Ag85A-membrane-bound protein. This finding encouraged the use of Apt to recognize the target pathogen. Thus, the proposed mechanism of the designed formulation for anti-TB treatment is based on the recognition and binding of Apt-functionalized niosome to the Ag85A-membrane-bound-proteins of Mtb. The Apt-PEGNio/EEP then passes through the Mtb cell wall and releases the EEP in the mycobacterial cytosol compartment. The released bioactive compounds of EEP possibly interact with growth-associated proteins and interfere with cell wall integrity, eventually causing a reduction in Mtb viability.

For further application in pulmonary deliver systems, the cytotoxicity of Apt-PEGNio/EEP was evaluated in alveolar macrophages. The results revealed the non-toxicity of this nano-formulation, which is in agreement with other studies using similar materials [24,53]. Importantly, the safety profiles of the constituents in the formulation were approved by the US FDA for internal use [42]. According to the results, this surface-functionalized niosomal vesicle with Apt could be a promising drug delivery system for Mtb targeted delivery of EEP. Notwithstanding, an understanding of the mode of action of this candidate tool in the infection event is still required.

## 5. Conclusions

In this work, we achieved the fabrication of the Apt-functionalized- niosome to improve the performance and enhance the therapeutic effects of EEP against Mtb. The niosome was loaded with bioactive EEP, then, its surface was modified with Ag85A Apt conjugation. As a targeting ligand of Apt, Apt-PEGNio/EEP specifically attached to Mtb and exhibited a supereminent anti-Mtb effect. Importantly, this nano-formulation was found to be non-toxic in vitro on the alveolar macrophage cells, meaning that it could be employed for further investigation. Our formulation will be proposed as a smart delivery device and might be a potential alternative therapeutic strategy to overcome the challenges of TB therapy.

## Figures and Tables

**Figure 1 nanomaterials-13-00269-f001:**
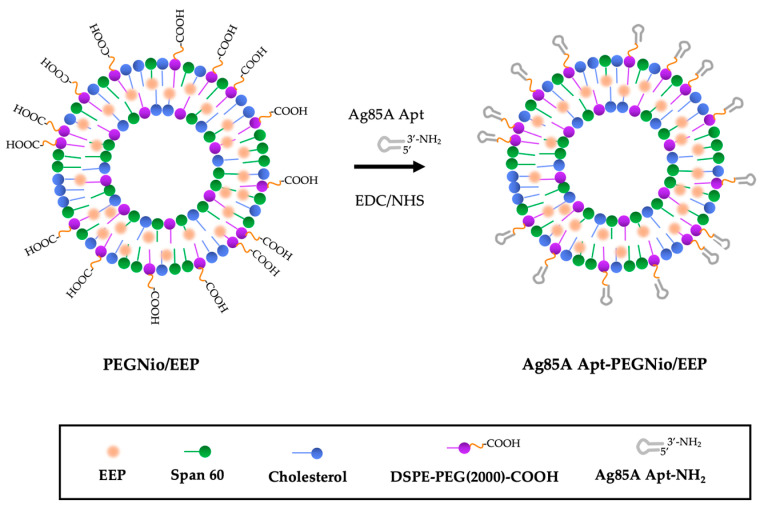
Schematic representation of the drug encapsulation and the bioconjugation process.

**Figure 2 nanomaterials-13-00269-f002:**
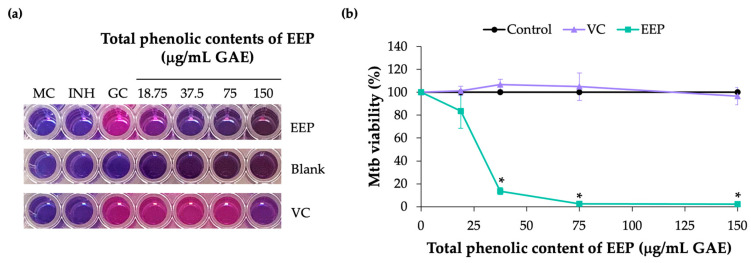
Inhibitory effect of EEP on Mtb. The viability of Mtb based on colorimetric reaction of resazurin dye was examined after treatment with EEP, compared to media control (MC) and growth control (GC). Isoniazid (INH) at 1 μg/mL was used as positive control for growth inhibition (**a**). Reduction of EEP treated Mtb was demonstrated in a comparison to growth control (Control) and vehicle control (VC) (**b**). All values are expressed as mean±SEM of three independent biological repeats. * *p* < 0.05, significant compared to VC.

**Figure 3 nanomaterials-13-00269-f003:**
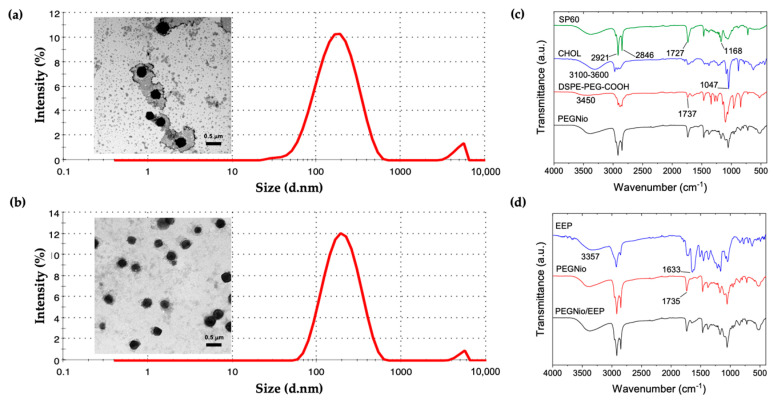
Characteristics of F2 formulation. Morphology and size distribution of PEGNio (**a**) and PEGNio/EEP (**b**) based on STEM. The scale bar represents 0.5 μm. FT−IR spectra of Span60 (SP60), CHOL, DSPE−PEG−COOH, PEGNio (**c**), EEP, PEGNio and PEGNio/EEP (**d**) were indicated.

**Figure 4 nanomaterials-13-00269-f004:**
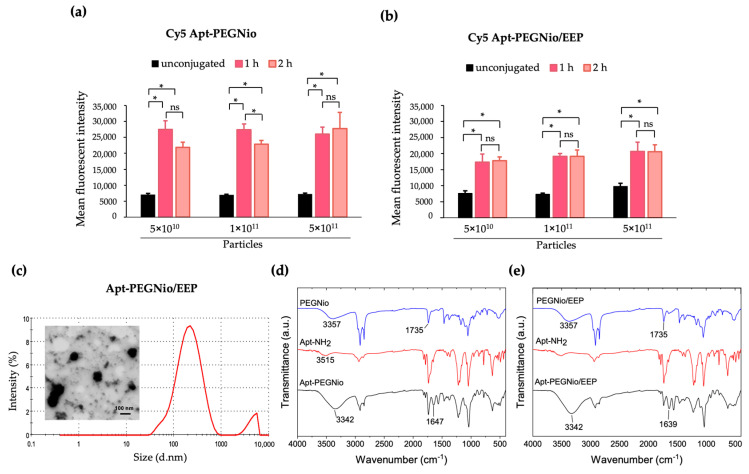
Conjugation of Apt to PEGNio/EEP. The attachment of Apt to PEGNio or PEGNio/EEP at different reaction time was determined by the fluorescent intensity of Ag85A Apt that labeled with Cy5 (**a**,**b**). The results were represented as the mean fluorescent intensity (MFI) of three independent experiments performed in triplicate. * *p* < 0.05 when compared to each group. ns: not significant. Morphological characteristic, size distribution (**c**) and FT−IR spectra of Apt-PEGNio (**d**) and Apt-PEGNio/EEP (**e**) were illustrated.

**Figure 5 nanomaterials-13-00269-f005:**
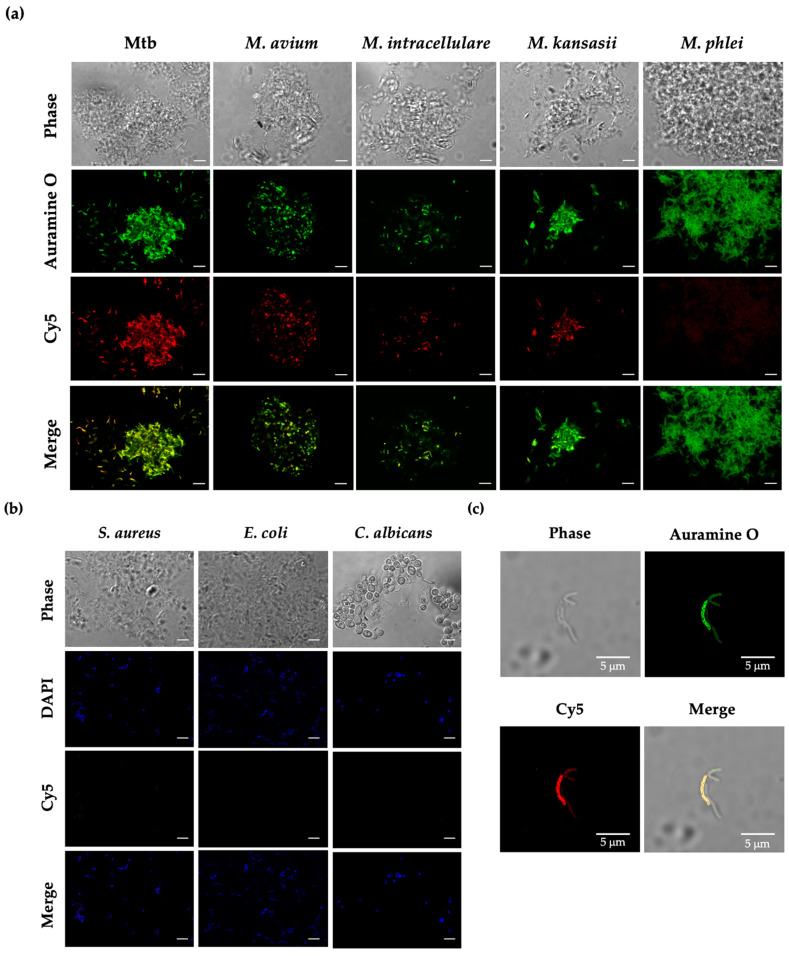
Binding of Apt-PEGNio/EEP to the microorganisms. Approximately 5 × 10^7^ CFU of mycobacterium (**a**), *S. aureus*, *E. coli* and *C. albicans* (**b**) were fixed with 4% paraformaldehyde and incubated with 1 × 10^12^ particles of Cy5 Apt-PEGNio/EEP (Red) for 1 h. The bacterial cells were washed and stained with Auramine O (Green) for Mycobacterium or DAPI (blue) for *S. aureus*, *E. coli* and *C. albicans*. Accumulation of Apt-PEGNio/EEP on Mtb cell wall was shown (**c**). Images were taken using a CLSM. The scale bar represents 5 μm.

**Figure 6 nanomaterials-13-00269-f006:**
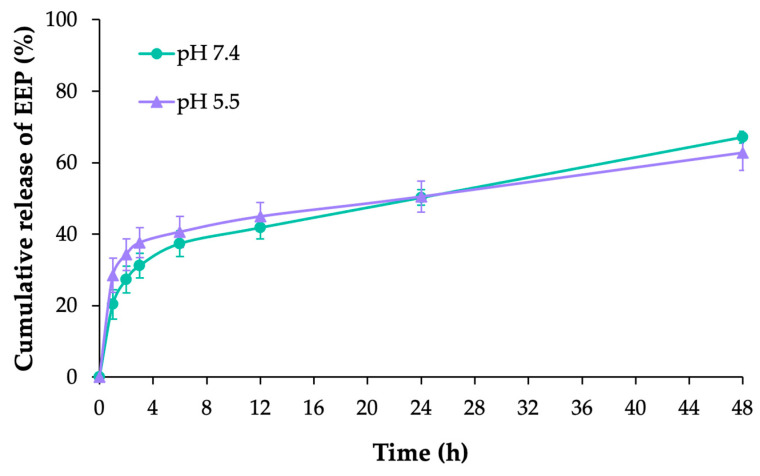
In vitro cumulative release of EEP from the Apt-PEGNio/EEP. The data are represented as the means ± SEM of three independent trials.

**Figure 7 nanomaterials-13-00269-f007:**
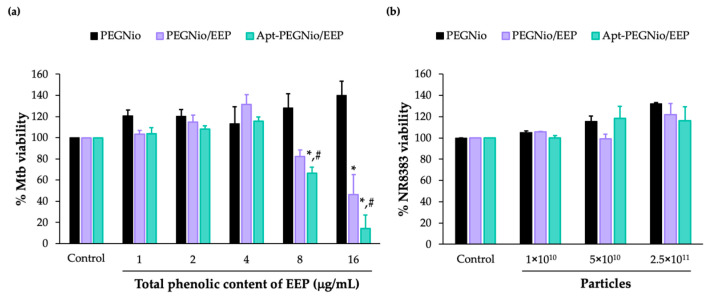
Evaluation of anti-Mtb activity and cytotoxicity of the nano-formulations. Reduction of Mtb viability (**a**) and non-toxicity on NR8383 cell (**b**) after treatment with the PEGNio, PEGNio/EEP, and Apt-PEGNio/EEP were investigated by REMA assay. All values are expressed as mean ± SEM of three independent biological repeats. * *p* < 0.05, significant compared to PEGNio, ^#^ *p* < 0.05, significant compared to PEGNio/EEP.

**Table 1 nanomaterials-13-00269-t001:** Compositions of different NP formulations.

Formulations	SP 60:CHOL:DSPE-PEG (mM)	SP 60 (mg)	CHOL (mg)	DSPE-PEG (mg)	Total Phenolic Content of EEP (mg/mL GAE)
F1	7:3:0.1	7.5	2.88	0.75	0.175
F2	5:5:0.1	5.4	4.75	0.75	
F3	3:7:0.1	3.25	6.75	0.75	

Abbreviations: SP 60, Span 60; CHOL, cholesterol; DSPE-PEG-COOH, DSPE-PEG(2000)-COOH.

**Table 2 nanomaterials-13-00269-t002:** Particle Size, Polydispersity Index (PDI), ζ Potential (ZP), Entrapment Efficiency (EE) and Total Phenolic Contents of Nano-formulations.

Formulations	PEGNio			PEGNio/EEP				
	Size (nm)	PDI	ZP (mV)	Size (nm)	PDI	ZP (mV)	% EE	Total Phenolic Contents of EEP (mg/mL GAE)
F1	222.51 ±11.75	0.48 ±0.05	−16.33 ±1.08	266.87 ±13.92	0.44 ±0.02	−20.48 ±3.10	38.83 ±0.66	0.06 ±0.004
F2	161.13 ±6.09	0.31 ±0.05	−20.69 ±1.26	176.14 ±13.88	0.21 ±0.04	−21.86 ±0.36	80.38 *^,#^ ±7.58	0.14 ±0.016
F3	146.16 ±5.34	0.26 ±0.02	−26.21 ±9.70	253.91 ±63.22	0.20 ±0.06	−24.06 ±4.08	41.69 ±6.49	0.055 ±0.024

* *p* < 0.05, significant of F2 compared to F1, ^#^ *p* < 0.05, significant of F2 compared to F3.

## Data Availability

Not applicable.

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
