# Peer review of "Nano-Delivery System of Ethanolic Extract of Propolis Targeting Mycobacterium tuberculosis via Aptamer-Modified-Niosomes"

_nanomaterials, 2023, doi:10.3390/nano13020269_

Round 1
Reviewer 1 Report
Suggested changes:
1. Line 30, “physicochemical characterization of PEGNio/EEP indicated approximately -20 mV of zeta potential, 180 nm of spherical 29 nanoparticles, 80% of entrapment efficiency, and the sustained-release manner”. I would recommend using the word sustained-release profile instead.
2. I would recommend not using words such as “outstandingly” especially at the start of a sentence in line 34.
3. Inconsistency in completing sentence (eg. Line 64, 68, 77). Highly recommend proofreading the manuscript.
4. I would strongly recommend for English grammatically corrections (Line 84, 306)
5. Could the authors please comment on the negative zeta potential of ~ -20 mV and effect of nucleases on the delivery system.
6. Could the authors please explain the reduction in Mean fluorescence intensity between 4a and 4b. Why do the groups encapsulating EEP show lower MFI than Apt-PEGNio group.
7. Could the authors please provide appropriate controls for binding of Apt-PEGNio/EEP to Mtb (Figure 5), maybe Apt-PEGNio or PEGNio/EEP?
8. Did the authors investigate the release profile at two different pHs (7.4 and 5.4) for other control groups, such as PEGNio/EEP? Does coupling with aptamer have any effect on the release profile?
Author Response
1. Line 30, “physicochemical characterization of PEGNio/EEP indicated approximately -20 mV of zeta potential, 180 nm of spherical nanoparticles, 80% of entrapment efficiency, and the sustained-release manner”. I would recommend using the word sustained-release profile instead.
Response: As reviewer’s suggestion, we have replaced the word “sustained-release manner” with “sustained release profile” in the revised manuscript at page 1, line 30.
2. I would recommend not using words such as “outstandingly” especially at the start of a sentence in line 34.
Response: As reviewer’s suggestion, we have deleted the word “outstandingly” from the sentence at page 1, line 34 of the abstract section in the revised manuscript.
3. Inconsistency in completing sentence (eg. Line 64, 68, 77). Highly recommend proofreading the manuscript.
Response: As reviewer’s suggestion, we have re-written the sentences in the revised manuscript at page 3, line 72-80.
4. I would strongly recommend for English grammatically corrections (Line 84, 306)
Response: As reviewer’s suggestion, we have re-written the sentences in the revised manuscript at page 3, line 94 and page 9, line 333.
5. Could the authors please comment on the negative zeta potential of ~ -20 mV and effect of nucleases on the delivery system.
Response: 1) The presence of charge on the surface of nanoparticles produces a repulsive force between the particles resulting in the physical stability. Generally, the zeta potential (ZP) values other than -30 mV to +30 mV are recommended (Joseph and Singhvi, 2019). In this study, the ZP of the optimized formulation are approximately -20 mV indicating moderate values. To improve its physical stability, the ZP values need to be improved by using the charge inducing agent in the formulation.
2) According to our formulation system, the niosome was coupled to the DNA aptamer without the nuclease resistant activity. Since, Mtb was reported to secrete the extracellular nuclease (Dang et al., 2016) and the nuclease can be from human serum. Therefore, it might be possible that the nuclease can affect the integrity of Apt. However, the Ag85A aptamer-based aptasensor assay was tested in human serum samples. The result showed this aptasensor can detect the Ag85A protein implying that nuclease in the human serum did not affect the binding activity of Ag85A aptamer (Ansari et al., 2018).
6. Could the authors please explain the reduction in Mean fluorescence intensity between 4a and 4b. Why do the groups encapsulating EEP show lower MFI than Apt-PEGNio group.
Response: From the structure of functionalized niosome as shown in Figure 1 and the fact that EEP contains polyphenol compounds. Since polyphenol compounds was described to show fluorescence interference (Dirimanov S. and Hogger P., 2020). Thus, the emission of Cy5 intensity at the 5’ end of Apt-PEGNio/EEP groups when compared to the Apt-PEGNio was reduced, as a result from the EEP.
7. Could the authors please provide appropriate controls for binding of Apt-PEGNio/EEP to Mtb (Figure 5), maybe Apt-PEGNio or PEGNio/EEP?
Response: The aim of this study was to ensure the specific binding of Ag85A Apt to Ag85A of Mtb. Thus, Apt-PEGNio/EEP or Apt-PEGNio can be used to evaluate the binding ability to the strains that can produce Ag85 complex. PEGNio/EEP was considered as an inappropriate control for the binding test due to it was not coupled with Ag85A Apt.
8. Did the authors investigate the release profile at two different pHs (7.4 and 5.4) for other control groups, such as PEGNio/EEP? Does coupling with aptamer have any effect on the release profile?
Response: The release profile of non-functionalized formulations or PEGNio/EEP was also investigated at pH7.4 and 5.5 in the same condition to Apt-PEGNio/EEP. The result demonstrated the sustained-release profile of all formulation even different pHs as shown in Figure 1 (in the revised manuscript). In this study, we focused on the pathological condition of this disease which presented an acidic pH (Gouzy et al., 2021) that showed no differences in the release profile of niosome coupling with and without Apt.
In addition, the English language editing was certified by MDPI. The certificate is shown below.
References
- Joseph, E.; Singhvi, G. Multifunctional nanocrystals for cancer therapy: a potential nanocarrier. In: Grumezescu AM, editor. Nanomaterials for Drug Delivery and Therapy. William and Andrew: Applied Science Publisher; 2019, 91–116.
- Dang, G.; Cao, J.; Cui, Y.; Song, N.; Chen, L.; Pang, H.; Lui, S. Characterization of Rv0888, a novel extracellular nuclease from Mycobacterium tuberculosis. Scientific Reports 2016, 6, 19033.
- Ansari, N.; Ghazvini, K.; Ramezani, M.; Shahdordizadeh, M.; Yazdian-Robati, R.; Abnous, K.; Taghdisi, S.M. Selection of DNA aptamers against Mycobacterium tuberculosis Ag85A, and its application in a graphene oxide-based fluorometric assay. Mikrochim Acta 2017, 185, 21.
- Dirimanov, S.; Hogger, P. Fluorescence interference of polyphenols in assays screening for dipeptidyl peptidase IV inhibitory activity. Food Frontiers 2020, 1, 484-492.
- Gouzy, A.; Healy, C.; Black, K.A.; Rhee, K.Y.; Ehrt, S. Growth of Mycobacterium tuberculosis at acidic pH depends on lipid assimilation and is accompanied by reduced GAPDH activity. PNAS 2021, 118, e2024571118.

Reviewer 2 Report
The authors (Tragoolpua et al.) presented the manuscript “Nano-delivery system of Ethanolic Extract of Propolis Targeting Mycobacterium tuberculosis via Aptamer Modified-Niosomes”. In quest of that authors fabricated niosome-based drug delivery platform for ethanolic extract of propolis (EEP) using a thin film hydration method and further surface modification with Ag85A aptamer (Apt-PEGNio/EEP) to enhance the therapeutic effects of EEP against Mtb. The study also supported that the targeting ligand of Apt, Apt-PEGNio/EEP acts as an antimycobacterial nanoparticle and might be a promising candidate for innovative targeted treatment of Mtb. Further application of this smart nano-delivery system may overcome the therapeutic challenges of TB therapy. Overall, the manuscript is organized, and the writing is readable. The methodology adopted is quite satisfactory. However, there are several minor deficiencies in the presented manuscript. I would recommend the publication of the manuscript after the authors address the following points.
1. One figure might be added to clearly present the concept. This figure may be merged with graphical abstract.
2. Kindly explain the mechanistic elaboration of the designed formulation for anti-TB treatment in the discussion part.
3. Novelty is questionable. Kindly explain this.
4. Please explain the rationale for the selection of Niosomes as a nanocarrier
5. Please explain the mechanism of propolis extract is useful for anti-TB potential. The author must give more emphasis on this part to clearly explain the scientifically.
6. Reviewers encourage to authors if they add the gaussian curve of the size distribution (PSD) of optimized formulations obtained from DLS in the revised manuscript.
7. Was the physical stability of the optimized formulation checked? Please explain.
8. Authors should compare the results of table 2 by statistical program and include them in the manuscript.
9. The significant digits in tables and manuscripts should be corrected.
10. Authors should elaborate more about the future perspective of this formulation in the conclusion section.
11. Authors should double-check the grammatical and punctuation errors in the manuscript.
Author Response
1. One figure might be added to clearly present the concept. This figure may be merged with graphical abstract.
Response: As reviewer’s suggestion, we have provided the graphical abstract in the revised manuscript at page 2.
2. Kindly explain the mechanistic elaboration of the designed formulation for anti-TB treatment in the discussion part.
Response: we have provided the mechanism of the optimized formulation in the discussion part in the revised manuscript at page 16, line 568-574 as shown below. “The proposed mechanism of the designed formulation for anti-TB treatment is based on the recognition and binding of Apt-functionalized niosome to the Ag85A membrane bound proteins of Mtb. The Apt-PEGNio/EEP then passes through the Mtb cell wall and releases the EEP in the mycobacterial cytosol compartment. The released bioactive compounds of EEP possibly interact with growth-associated proteins and interfere with cell wall integrity, eventually causing a reduction in Mtb viability.”
3. Novelty is questionable. Kindly explain this.
Response: This is the first report to utilize the Apt-functionalized niosomal carriers for the efficient delivery of EEP. The optimized formulation decorating with Apt not only enhanced the specific interaction to the targeting ligand, but also increased the internalization of the bioactive agents to Mtb. Our formulation will be proposed as a smart delivery device and might be a potential alternative therapeutic strategy in treatment of pulmonary tuberculosis.
4. Please explain the rationale for the selection of Niosomes as a nanocarrier
Response: Since, systemic administration of anti-TB agents has a limitation to penetrate the alveolar macrophages. Moreover, the lung fluid and pulmonary surfactant also act as a biological barrier to antimicrobials delivery in the lungs. Hence, niosomes system-based drug delivery was chosen to enhance penetration efficiency for further application in pulmonary tuberculosis treatment.
5. Please explain the mechanism of propolis extract is useful for anti-TB potential. The author must give more emphasis on this part to clearly explain the scientifically.
Response: The anti-bacterial mechanism of propolis is based on the function of its polyphenolic constituents that possibly affects cytoplasmic membrane function, nucleic acid synthesis, metabolism, or formation of biofilm (Sawicki et al., 2022). Moreover, Ali et al., 2018 reported that various known constituents in propolis such as flavonoids, terpenoids, simple phenolics, pterocarpan, phenylethanoid derivative, and stilbenes showed the interaction with the key proteins related Mtb growth and many essential mycobacterial pathways such as biosynthesis of cell wall and co-factors, and signal transduction.
6. Reviewers encourage to authors if they add the gaussian curve of the size distribution (PSD) of optimized formulations obtained from DLS in the revised manuscript.
Response: As reviewer’s suggestion, we have provided the size distribution curve of the optimized formulation obtained from DLS in Figure 3 and Figure 4 of the revised manuscript at page 9 and 10, respectively.
7. Was the physical stability of the optimized formulation checked? Please explain.
Response: We agree with your suggestion. The physical stability of the optimized formulation did not check in this study. From this work, powder-derived niosome will be performed and the long-term storage stability will be further checked.
8. Authors should compare the results of table 2 by statistical program and include them in the manuscript.
Response: We have compared the %EE of three formulations using one-way analysis of variance. *p<0.05, significant of F2 compared to F1, #p<0.05, significant of F2 compared to F3. However, no statistical differences of average particle size, PdI, and ZP between each group as shown in Table 2 of the revised manuscript at page 8.
9. The significant digits in tables and manuscripts should be corrected.
Response: As reviewer’s suggestion, we have corrected the digit number in Table 2, at page 8 in the revised manuscript.
10. Authors should elaborate more about the future perspective of this formulation in the conclusion section.
Response: As reviewer’s suggestion, we have added the following sentence in the conclusion section of the revised manuscript at page 16, line 592-594.
“Our formulation will be proposed as a smart delivery device and might be a potential alternative therapeutic strategy to overcome the challenges of TB therapy.”
11. Authors should double-check the grammatical and punctuation errors in the manuscript.
Response: As reviewer’s suggestion, we have checked the grammatical and punction errors in the revised manuscript. The English language editing ID: english-57958 was certified by MDPI.
References
- Sawicki, R.; Widelski, J.; Okinczyc, P.; Truszkiewicz, W.; Glous, J.; Sieniawska, E. Exposure to Nepalese propolis alters the metabolic state of Mycobacterium tuberculosis. Front Microbiol 2022, 13, 929476.
- Ali, M.T.; Blicharska, N.; Shilpi, J.A.; Seidel, V. Investigation of the anti-TB potential of selected propolis constituents using a molecular docking approach. Sci Rep 2018, 8,
